# Optimal Inspired Fraction of Oxygen in the Delivery Room for Preterm Infants

**DOI:** 10.3390/children6020029

**Published:** 2019-02-19

**Authors:** Inmaculada Lara-Cantón, Alvaro Solaz, Anna Parra-Llorca, Ana García-Robles, Máximo Vento

**Affiliations:** 1Neonatal Research Group, Health Research Institute La Fe, Avenida Fernando Abril Martorell 106, 46026 Valencia, Spain; Inmb612@gmail.com (I.L.-C.); alvarosogar@gmail.com (A.S.); annaparrallorca@gmail.com (A.P.-L.); garcia.anarob@gmail.com (A.G.-R.); 2Division of Neonatology, University and Polytechnic Hospital La Fe, Avenida Fernando Abril Martorell 106, 46026 Valencia, Spain

**Keywords:** prematurity, resuscitation, delivery room, inspired fraction of oxygen, pulseoximetry, oxygen saturation range

## Abstract

Postnatal adaptation of preterm infants entails a series of difficulties among which the immaturity of the respiratory system is the most vital. To overcome respiratory insufficiency, caregivers attending in the delivery room use positive pressure ventilation and oxygen. A body of evidence in relation of oxygen management in the delivery room has been accumulated in recent years; however, the optimal initial inspired fraction of oxygen, the time to achieve specific oxygen saturation targets, and oxygen titration have not been yet clearly established. The aim of this review is to update the reader by critically analyzing the most relevant literature.

## 1. Introduction

Lung cytoarchitecture, thoracic cage structure, and muscular strength only mature late in gestation [1]. Moreover, metabolic components that essentially contribute to an effective respiration, such as surfactant and antioxidant enzymatic and non-enzymatic defenses, are not produced in sufficient quantity until the last weeks of gestation [2]. Consequently, preterm infants, and especially very preterm infants with gestational ages below 32 weeks, frequently undergo respiratory difficulties immediately after birth. Generalized atelectasis, inability to establish a functional residual capacity, hypoxemia, hypercapnia, and increased work of breathing are the clinical characteristics that describe the respiratory failure of preterm infants when initiating air breathing. Under these circumstances, prenatal interventions such as the administration to the mother of antenatal steroids, and postnatal ventilation and oxygen supplementation to the newborn preterm infants render essential to overcome this dramatic situation [3]. The benefits of antenatal steroids and postnatal non-invasive ventilation in the delivery room (DR) have been widely acknowledged. Nonetheless, despite the generalized acceptance of oxygen as the most important drug for preterm resuscitation, there are still relevant aspects in the management of oxygen that are yet to be answered. Hence, the preferred initial inspired fraction of oxygen (FiO_2_), the oxygen saturation (SpO_2_) target ranges along the first minutes after birth, and how to perform the adjustments to increase or decrease FiO_2_ along stabilization still require further evidence. There is an inherent risk to using inappropriate oxygen concentrations in very preterm infants. Oxygen in excess causing hyperoxemia leads to an enhanced production of oxygen free radicals, oxidative stress, and tissue damage in preterm babies with an immature antioxidant defense system [2]. On the contrary, hypoxemia especially when combined with bradycardia significantly enhances the risk for intraventricular hemorrhage (IVH) and death. Both these situations increase mortality and/or short-long term morbidities in survivors [4].

The aim in the present review article is to critically analyze the most relevant and recent literature concerning the use of oxygen in the DR to help neonatologists improving the management of preterm infants during postnatal stabilization.

## 2. Oxygen in the Fetal to Neonatal Transition

Fetal life elapses in a low oxygen environment as compared to the extrauterine environment. The arterial partial pressure of oxygen in utero is about 25–30 mmHg as compared to 80–90 mmHg in the mother [5]. Of note, the most oxygenated fetal blood is directed through circulatory shunts to the brain and myocardium which are the two most oxygen demanding tissues [6]. Initiation of breathing immediately after birth triggers profound cardiorespiratory and metabolic changes. Lung expansion with the initial inspiratory movements and extrusion of lung fluid from the respiratory airways and alveoli to the interstitium contributes to dilatation of the lung vasculature, drop of the vascular resistance, closure of the intra-and-extra-cardiac shunting, and redirecting of the ventricular output to the lungs, where it gets oxygenated [7]. PaO_2_ rises abruptly to 70–80 mmHg in the first 5–10 min after birth. Arterial oxygen saturation (SpO_2_) reflects the percentage of hemoglobin that is saturated with oxygen. Under physiologic circumstances SpO_2_ range oscillates between 95% and 100% in newborn infants [7].

What is the timing for postnatal SpO_2_ stabilization? Dawson et al. merged three different data bases from term and preterm newborn infants who did not need resuscitation or oxygen supplementation upon stabilization. With these data they assembled an oxygen saturation range graph with centiles for term and preterm babies for the first 10 min after birth [8]. Reference ranges for term infants have been adopted by international guidelines to establish target SpO_2_ recommendations minute by minute. Thus, recommended range for SpO_2_ at 1 min 60–65%, 2 min 65–70%, 3 min 70–75%, 4 min 75–80%, 5 min 80–85% and 10 min 85–95% [9]. However, the reference ranges for preterm infants were based on a population mainly composed by late preterm infant (33^+6^ to 36^+6^ weeks gestation) with very little representation of very preterm infants [10]. Interestingly, very preterm infants using positive pressure ventilation with air achieved significantly earlier Dawson’s nomogram SpO_2_ targets introducing an additional confounder for the clinician [11].

Oei et al. launched an international survey to determine current clinical practice and opinions regarding FiO_2_ and SpO_2_ targets for DR resuscitation of very preterm infants [12]. The majority of neonatologists (77%) who participated in the survey would target SpO_2_ between the 10th and 50th percentiles of the reference range for full-term infants [8], and would choose to start with an initial FiO_2_ 0.3; however, most of the interviewed neonatologists acknowledged that evidence was lacking and further research is needed [12].

The generalization of delayed cord clamping in the delivery room (DR) in preterm infants and clinical trials that are investigating the initiation of ventilation before cord clamping are adding new variables that may influence the oxygen target ranges in preterm infants upon stabilization [13,14].

## 3. Initial FiO_2_ in the Delivery Room

In 2015, the International Liaison Committee on Resuscitation [15], American Heart Association (AHA) [9], and the European Resuscitation Council (ERC) [16], strongly recommended to initiate resuscitation of preterm infants with an FiO_2_ between 0.21 and 0.30, although evidence showed that no differences in clinical outcomes were present when initiating resuscitation either with lower (<0.3) or higher (>0.6) initial FiO_2_. However, criticism arose as neither of the recommendations included relevant studies that reported an increased mortality and morbidity, especially in extremely preterm infants (≤28 weeks’ gestation), when stabilization was initiated with air [17]. Hence, while a meta-analysis published in 2014 by Saugstad et al. [18], had shown that preterm babies ≤32 weeks’ gestation resuscitated with lower (0.21–0.3) versus higher (0.6–1.0) initial FiO_2_ showed no differences in morbidity but exhibited an almost significant trend towards reduced mortality [18], both the TORPIDO trial from Australia/New Zealand/Malaysia [19] and the Neonatal Network in Canada [20] reported increased mortality in very preterm babies initially stabilized with air. The results of the TORPIDO trial [19] showed in 289 preterm infants <32 weeks’ gestation that the mortality in a subgroup of babies <29 weeks’ gestation (16.2%) was significantly increased as compared with the mortality (6%) in the pure oxygen group (*p* = 0.013). Rabi et al. [20], in a retrospective study, compared the results of babies ≤27 weeks’ gestation from the Canadian Neonatal Network before and after 2006 when the initial FiO_2_ was changed from 1.0 to <1.0 with the primary outcome being severe neurologic injury or death. Adjusted odds ratio for the primary outcome was significantly higher in the lower oxygen group (AOR 1.36; 95% CI 1.11–1.66) and in those resuscitated with RA (AOR 1.33; 95% CI 1.04–1.69), when compared with 100% oxygen [20]. These studies [19,20] underpinned the necessity of performing in depth analysis of the available information and the stringent necessity to launch adequately designed and powered studies to overcome the uncertainty about the use of oxygen in the DR in these very vulnerable infants.

Welsford et al. recently published a systematic review and meta-analysis summarizing the available evidence for the FiO_2_ used to initiate resuscitation of preterm infants who require positive pressure ventilation in the DR [21]. Data pertained to 10 randomized/quasi randomized controlled and 4 retrospective observational cohort studies—including a total of 5697 preterm infants <35 weeks’ gestation. The use of higher (≥0.5) or lower (<0.5) initial FiO_2_ did not report benefit or harm in the primary outcome of short-term mortality (risk ratio (RR) 0.83, 95% confidence interval (CI) 0.50–1.37), long-term mortality, neurodevelopmental impairment, or other morbidities [21]. The results of this study should be interpreted cautiously. The study encompasses from 1980 to 2018 and includes a wide range of different designs and gestational ages. Finally, it also has included studies performed with enormous differences in the technology employed in the DR, especially pulse oximetry and air/oxygen blender. The authors confirmed that the study has important limitations especially due to the risk of bias and imprecision and conclude that the ideal initial FiO_2_ for preterm infants it yet unknown despite the fact that many of these infants will require oxygen supplementation upon stabilization in the DR [21].

Almost simultaneously, a Cochrane systematic review has analyzed lower (<0.4) vs. higher (≥0.4) initial FiO_2_ titrated to target oxygen saturations during resuscitation of preterm infants [22]. The search included only randomized controlled trials and the study period was also shorter (2008–2016). Interestingly, in each of these studies SpO_2_ and heart rate were monitored and FiO_2_ titrated according to targeted saturations. No differences in mortality between both groups could be assessed (RR 1.05, 95% CI 0.68–1.63). However, the quality of evidence was graded as low due to risk of bias and imprecision [22]. No differences were shown in other relevant outcomes such as ROP, PVL, IVH, NEC, BPD or PAD. The authors concluded that there is uncertainty as to whether the use of higher or lower initial FiO_2_ affects short or long-term outcomes in preterm infants [22]. Clinical studies have shown that very preterm infants have difficulty achieving adequate oxygenation in the first minutes after birth when supplemented with room air. In a preterm lamb model of asphyxia/resuscitation using FiO_2_, 0.21 did not reduce pulmonary vascular resistance as compared to FiO_2_ of 1.0 [23]. On the contrary, the use of 100% oxygen caused hyperoxemia detected in the carotid arteries but did not cause cerebral vasoconstriction; however, cerebral oxygen extraction did not differ between groups [23]. Very preterm infants need higher FiO_2_ than term infants to dilate the pulmonary vessels and avoid hypoxemia and/or bradycardia; notwithstanding, the risk of cerebral hyperoxia and subsequent brain damage should be avoided using intermediate inspired fractions of oxygen, closely titrating SpO_2_ [23]. The use of near infrared spectroscopy (NIRS) in the delivery room allows to monitor evolving cerebral regional oxygen saturation and fraction of oxygen extraction rapidly and accurately after birth. Brain oxygenation measured by NIRS reaches a plateau much faster than arterial oxygen saturation. Cerebral tissue oxygenation relies not only in arterial oxygen saturation but also in cerebral blood flow, hemoglobin content and oxygen consumption. Thus, with the information provided by NIRS oxygen targeting especially in the first minutes after birth can be more accurately achieved and cerebral hyper-or-hypoxia more easily avoided [24,25].

## 4. Target Oxygen Saturation during Stabilization

AHA 2015 guidelines recommend that FiO_2_ should be titrated according to pre-established saturation targets from spontaneously breathing, full-term infants [9]. Despite this recommendation, SpO_2_ is not routinely registered as complementary information to the Apgar score. A prospective observational study, however, showed that reflecting objective information during postnatal stabilization, the so-called expanded Apgar score, would not only better reflect the process of adaptation but also it would better predict early and late clinical outcomes [26]. In individual patient data from 768 infants <32 weeks’ gestation enrolled in 8 RCTs, infants who did not reach a SpO_2_ 80% by 5 min were more likely to develop a severe intraventricular hemorrhage (OR 2.04, 95% CI 1.01–4.11) and die (OR 4,.57, 95% CI 1.62–13.98) compared to infants with SpO_2_ >80% [27]. Infants were also less likely to reach SpO_2_ 80% if resuscitation was commenced with lower (≤0.3) FiO_2_ instead of higher (>0.6) FiO_2_ (OR 2.63, 95% CI 1.21–5.74) were more premature and had lower birth weight [27]. Similarly, Thamrin et al. [28] reported the results of a follow up study at 2–3 years corrected age of preterm babies born at <32 weeks’ gestation and resuscitated with FiO_2_ or either 0.21 or 1.0. They found no differences in Bayley Scales independently of the initial FiO_2_ employed. Interestingly, in a post hoc exploratory analysis of the entire cohort, babies who did not reach SpO_2_ 80% at five minutes after birth were more likely to die or have neurodevelopmental impairment (OR, 1.85; 95% CI 1.07–3.2; *P* = 0.03). From the results of these studies [27,28], it may be deduced that more than starting with a specific FiO_2_, performing an adequate ventilation and oxygen titration that allows the patient reaching a specific SpO_2_ at a given time point after birth has a notable influence on major outcomes. Inability to do so could be informing about prenatal complications in utero or inadequate resuscitation. Thus, SpO_2_ at 5 min after birth should be registered together with the 5 min Apgar score. Moreover, it should have an influence on the pace of FiO_2_ adjustments to reach the SpO_2_ target. Babies who do not reach 80–85% saturation around 5 min should be considered at high risk of severe complications and must be closely monitored.

## 5. Heart Rate

Myocardial function is indispensable for an adequate tissue perfusion and oxygenation. Cardiac output is the result of stroke volume and heart rate. Cardiac function is highly dependent on aerobic metabolism. Coronary blood flow is regulated by arterial blood oxygen content. Therefore, under hypoxemic conditions myocardium suffers hypoperfusion and energy exhaustion which lead to bradycardia and generalized hypoperfusion and hypoxemia [29]. Heart rate is therefore the most critical sign for an adequate response to resuscitation maneuvers. Monitoring heart rate (HR) even if SpO_2_ monitoring is not available may be highly informative. The Bradyprem study [30] is a multicenter retrospective study that retrieved data from 596 infants <32 weeks’ gestation from 8 RCT of higher (>0.6) vs. lower (≤0.3) FiO_2_. The study aimed to determine if there was an association between bradycardia and neonatal morbidity and mortality during stabilization in preterm infants during the first 10 min after birth. In addition, the study also aimed to conclude if there was an interaction between prolonged bradycardia and low SpO_2_ during resuscitation [30]. The primary outcome was in-hospital mortality and secondary outcomes were intraventricular hemorrhage (IVH, ≥grade 3), bronchopulmonary dysplasia (BPD) and retinopathy of prematurity (ROP, ≥grade 3). Results showed that 38% of babies in the DR experienced prolonged bradycardia (HR <100 bpm for >2 minutes). These babies were more likely to have 5 min SpO_2_ <80%, and more likely to die and/or to have IVH. Duration of bradycardia was associated with higher mortality even after adjusting for confounders such as gestational age, birth weight, gender, antenatal steroids, SpO_2_ at 5 min <80% and individual study variability. An accurate and continuous measurement of HR during neonatal resuscitation is critical and evaluation of long-term outcomes of patients who experienced prolonged bradycardia in the first 10 min after birth are paramount [30].

## 6. Follow-Up

The long-term consequences of hypoxia and/or hyperoxia during postnatal stabilization have only been recently studied. As shown in recent meta-analyses [21,22] follow-up studies of babies resuscitated with different initial FiO_2_ rates are relatively scarce. However, it is well-known that both hypoxia and hyperoxia can directly cause neuronal necrosis and/or activate pro-inflammatory and pro-apoptotic pathways causing deleterious effects on the developing brain [31]. Soraisham et al. [32] performed a retrospective cohort study that included 1509 preterm infants <29 weeks’ gestation resuscitated either with room air, intermediate oxygen concentration or 100% oxygen with the primary outcome being death or neurodevelopmental impairment (NDI). The results showed no differences for the composite outcome of death or severe NDI between the groups. However, the odds of severe NDI among survivors were significantly higher in infants that received 100% oxygen as compared to room air (aOR 1.57, 95% CI 1.05, 2.35) indicating that the use of pure oxygen could be damaging for the central nervous system in those babies who survived the neonatal period [32]. Similarly, Kapadia et al. [33], in a retrospective observational study, compared the impact of changing FiO_2_ from 1.0 to 0.21 introduced in 2011 in the Neonatal Resuscitation Program of his institution upon neonatal morbidities, mortality and neurodevelopmental outcomes in preterm infants. The results showed that babies resuscitated with a low oxygen approach had lower oxygen exposure, spent fewer days on oxygen, and had lower odds of developing BPD. Mortality was no different between groups; however, survivors of the air group had greater motor composites scores on Bayley III Scales [33] confirming results obtained in Canada by Shoraisham et al. [32]. Finally, Boronat et al. [34] included a total of 253 surviving preterm infants <32 weeks’ gestation from two randomized, double blinded, international, multicenter clinical trials in which patients were assigned to higher (0.6 or 0.65) or lower (0.3) initial FiO_2_ in the DR. A total of 206 (81.4%) completed follow up at 24 months that included Bayley III Scales comprehensive scoring, visual acuity, neurosensory deafness, and language skills. No differences in survival or NDI were found between both groups [34]. Results from follow up studies show that using an initial FiO_2_ of 1.0 increases the chance of NDI in survivors. However, when using initial FiO_2_ of 0.6–0.65 as shown in Boronat’s paper [34] no differences in neurocognitive or sensorial outcomes were present.

From the results shown in the follow up studies it may be deduced that the oxygen load defined as the amount of pure oxygen expressed in mL/kg body weight supplied to preterm infants during resuscitation causes a long-term impact on NDI. Oxygen load is highly dependent on gestational age, type of birth and gender and especially on initial FiO_2_ and titration protocols [35]. Moreover, the amount of oxygen provided immediately after birth significantly influenced DNA methylation causing hypomethylation of a significant number of genes related with cell cycle progression, DNA repair, and oxidative stress [36]. We can only speculate if oxygen load is responsible at least partially for long-term clinical outcomes.

## 7. Clinical Practice

Evidence based knowledge and clinical practice do not always run in parallel. While the scientific approach has not yet determined the optimal oxygen inspired fraction and the ideal procedure to titrate oxygen in the DR, the clinical practice has highly inclined the plate towards a lower initial FiO_2_ during postnatal stabilization. Avoiding hyperoxia-derived oxidative stress and its consequences has had a higher impact than hypoxemia during the first minutes after birth which seems to be a transitory physiological status especially for very preterm infants especially if the baby keeps a HR >100 bpm after two minutes after birth. Thereafter, titrating FiO_2_ always avoiding SpO_2_ >96% is the experts’ recommendation to avoid damage caused by hyperoxia [37]. In a survey performed among 630 clinicians from 25 countries the majority (77%) would target SpO_2_ between the 10th and 50th percentiles values for term infants as shown in the AHA 2015 guidelines [9] and expressed their preference for the use of 0.3–0.4 or even lower initial FiO_2_ (0.21) for preterm infants in the DR. Of note, a substantial number of responders agreed that there was a gap of knowledge and further studies would need to be performed [12]. More recently, a review of the guidelines for oxygen use in the DR in 45 different countries showed a great variability regarding gestational ages, initial FiO_2_ and SpO_2_ targets [38]. Initial FiO_2_ recommendations ranged from 0.21–1.0; however, the most frequently recommended were 0.21–0.3 (38%), 0.3 (20%) and 0.21 (18%). Differences in target SpO_2_ at five minutes were notably different between countries. Therefore, while Scandinavian countries recommended 70%, other countries such as New Zealand or Australia recommended 90% and some countries did not make any recommendation [38]. Another relevant question is around the adherence to guidelines. As expressed by Farquhar et al., although clinicians rely on guidelines as a good educational tool they considered them too rigid for a specific patient and limit physician’s own clinical choice [39]. In addition, the permeation of national guidelines is also highly dependent on the clinical level of the hospital. Thus, care givers in smaller countryside hospitals tend to substantially delay the application of new standards of care especially if they imply the use of novel technical devices, such as air/oxygen blenders, pulse oximeters or T-piece resuscitators. Hence, countryside hospitals significantly delayed the use of oxygen blenders for FiO_2_ titration and the systematic use of pulse oximeters when attending preterm infants in the DR as compared with level III reference regional centers as shown in the survey performed in Spain by Iriondo et al. [40].

## 8. Seeking an Answer: Forthcoming Trials

Meta-analyses and systematic reviews performed in the last few years, as shown in Section 3 and Section 4, have not been able to shed light on the optimal use of oxygen in preterm infants in the delivery room. Striking differences in favor of initiating resuscitation with higher (even 100%) oxygen, intermediate (60%) or lower (21–30%) have been concluded. The small size, differences in gestational ages, and varied designs and primary objectives, among other things, have led to different conclusions generating confusion among clinicians. To overcome this conundrum, large multicenter trials of sufficient sample size powered to look at safety outcomes such as mortality and/or major neurodevelopmental impairment are needed. The HiLo trial (NCT03825835) aims to answer if higher (60%) or lower (30%) initial oxygen concentration has an impact on patient relevant outcomes in preterm infants of 23^0^ to 28^+6^ weeks’ gestation. The null hypothesis is that neither mortality or NDI at 18–24 months corrected age will be different independently of the initial FiO_2_ employed. The study has a cluster crossover design, unmasked randomized controlled trial comparing two oxygen concentration at initiation of resuscitation. Hence, NICUs instead of infants will be randomized to an initial FiO_2_’s (e.g., higher) until several babies are recruited, and then switched to the other FiO_2_ (i.e., lower). The analysis will be conducted using an “intention-to-treat” approach. Interestingly, SpO_2_ will be controlled at 3 min after birth and according to the oxygen saturation increased or decreased by 20% every 60 seconds to achieve ≥85% between 5–10 min of age. A second study, the Torpido 30/60 (ACTRN12618000879268) trial, compares initial FiO_2_ of 0.3 vs. 0.6. This is a phase III, randomized, 2 parallel arms, non-blinded trial in preterm infants 23^+0^ to 28^+6^ weeks’ gestation. The aim of this study is to compare short-and-long term outcomes being the primary outcome survival free of major brain injury at 36 weeks, and secondary outcomes mortality and major brain injury. Eligible babies will be randomized to FiO_2_ of 0.3 or 0.6 before birth. Importantly, target SpO_2_ at five minutes should be 80–85% and at 10 min and after 85–95%. Both, the HiLo and Torpido 30/60 have been powered to detect an absolute risk difference around 10% in the primary endpoint. Both these ongoing trials seek answering many of the question’s neonatologists have to face when confronted with the resuscitation of very preterm infants.

## 9. Conclusions

A body of evidence for the field of newborn oxygenation has been accumulated. However, there is increasing awareness that more reliable data are needed. The tendency towards a reduction in the amount of oxygen given to newborn infants, especially very preterm, upon stabilization in the delivery room is an inarguable fact. The benefit of this clinical tendency is still under debate. The need of very preterm infants for some amount of oxygen and the risk of not overcoming bradycardia and hypoxemia and blunt the stimulation of the respiratory center to open the glottis with deleterious consequences upon the myocardium and central nervous system should be a matter of great concern. Published data reveal that very preterm infants but especially <28 weeks’ gestation will need supplemental oxygen in the first minutes after birth to overcome the tendency towards hypoxemia. Strict control of heart rate is mandatory because persistence of bradycardia is an ominous sign and highly predictive of death or major morbidities. Saturation at five minutes should be included together with the 5-minute Apgar score in the electronic registry of every baby. In addition, follow up is recommended for babies who had prolonged bradycardia and desaturation during postnatal adaptation independently of gestational age. Finally, suggestions for the use of oxygen in the delivery room in the first ten minutes after birth as an adaptation of a recent review publication of experts in the field [31] is shown in Table 1.

## Figures and Tables

**Table 1 children-06-00029-t001:** Suggestions on how to supply oxygen in the delivery room to newly born infants.

Gestational Age	Initial FiO_2_	Target SpO_2_ at 5 min
<37 weeks	0.21	85–90%
33^+0^ to 36^+6^ weeks	0.21	85%
29^+0^ to 32^+6^ weeks	0.21-0.30*	80–85%
≤28 weeks	0.3	80%

Abbreviations: FiO_2_: inspired fraction of oxygen; SpO_2_: arterial oxygen saturation measure by pulseoximetry. Further suggestions: initial FiO_2_ should be decided after evaluation of immediate postnatal reactivity by the attending neonatologist. If no reactive choose the higher FiO_2_; FiO_2_ adjustments should aim to achieve targeted SpO_2_ and avoiding bradycardia (heart rate <100 bpm); Hear rate >2 min after birth should be >100 bpm.; Register SpO_2_ at 5 min together with the 5-minute Apgar score; Modified from Saugstad et al. *Pediatr. Res.*
**2018** [31].

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
