# Peer review of "Optimal Inspired Fraction of Oxygen in the Delivery Room for Preterm Infants"

_children, 2019, doi:10.3390/children6020029_

Round 1

Reviewer 1 Report

A crisp review of oxygen use in the delivery room during preterm infants by Lara-Cantón et al. from Vento's group.

The summary of the review highlights the fact that despite decades of research, the initial inspired oxygen use in a preterm infant remains

The authors have written a concise review based on current literature and practice with relevant literature. The review also suggests different levels of inspired oxygen based on gestational age.

The reviewer has very few comments about this well-written review. I recommend adding a few newer literature and suggesting different approaches to monitoring oxygenation in the delivery room, which could potentially benefit the readers/researchers in this field.

Please include comments/review about the following relevant literature that was recently published:

1)      Preterm Infant Outcomes after Randomization to Initial Resuscitation with FiO2 0.21 or 1.0. Thamrin & Oei et al. Published in J peds. Oct 2018. PMID: 30251639.

The study is a follow up at 2 years. Blinded assessments were conducted at 2-3 years corrected age with the Bayley Scales of Infant and Toddler Development, Third Edition or the Ages and Stages Questionnaire by intention to treat. Initial resuscitation of infants <32 weeks' gestation with initial FiO2 0.21 had no significant effect on death or NDI compared with initial FiO2 1.0.

In post hoc exploratory analyses in the whole cohort, children with a 5-minute blood oxygen saturation (SpO2) <80% were more likely to die or to have NDI (OR, 1.85; 95% CI, 1.07-3.2; P = .03).

The author's viewpoint and comments on this study could be valuable in this review.

2)      Effect of various inspired oxygen concentrations on pulmonary and systemic hemodynamics and oxygenation during resuscitation in a transitioning preterm model. Chandrasekharan & Lakshminruimha et al. Published in Ped Research. Nov 2018. PMID 29967523.

The study is in a preterm lamb model where they have used term lambs as controls. Initial resuscitation with 21% O2 followed by titration of O2 led to suboptimal pulmonary vascular transition at birth in preterm lambs. Ventilation with 100% O2 in preterm lambs caused hyperoxia but reduced PVR similar to term lambs on 21% O2.

The novel part of this study is not highlighted in their abstract.

a)       The difference in systemic and pulmonary blood pressures with 100% oxygen that is not evident with 21% is startling.

b)      The oxygen delivery was higher with high oxygen use, and so was the regional cerebral saturation but not oxygen extraction.

Could monitoring cerebral saturation be a better marker of oxygenation in the delivery room than peripheral saturation?

Would 5-minute Apgar give an idea of perfusion or should the perfusion be noted separately? 

The author’s viewpoint on these hemodynamic variables and alternate methods of monitoring oxygenation will also be valuable to this review.

3)      Cerebral Oxygen Saturation to Guide Oxygen Delivery in Preterm Neonates for the Immediate Transition after Birth: A 2-Center Randomized Controlled Pilot Feasibility Trial. Pichler & Schmölzer. Published J peds 2016. PMID: 26743498

Reduction of the burden of cerebral hypoxia during immediate transition and resuscitation after birth is feasible by crSO2 monitoring to guide respiratory and supplemental oxygen support.

Discussing the positives & negatives of the practical aspects of monitoring cerebral oxygen saturation in the delivery room would be valuable too.

Comments by section:

2. Oxygen in the fetal to neonatal transition: in this section the authors mention - ‘clinical trials that are investigating the initiation of ventilation before cord clamping’.

How could the research of sustained inflation in preterm infants have an impact?

3. Initial FiO2 in the delivery room: How should the oxygen be titrated in the delivery room? How soon and what should be the change while titrating oxygen? Could the authors comment on this? Especially when there is a delay between titration and the actual oxygen concentration delivered to the lungs.

AHA mentions starting the initial oxygen anywhere from 21-30 up to 60% but not greater than 65%, while the authors have mentioned in the table (at the end of the manuscript) to start at not more than 40%. Please comment.

4. Target oxygen saturation during stabilization: Should we abolish minute-to-minute saturation targets and focus on a broader range in the first 5 minutes? ERC, AHA, ANZ, Finland, South Africa, Sweden all differ (this has been mentioned in the clinical practice section). Should there be a universal common target at 5 min? Is there an advantage/disadvantage in monitoring targets every minute?

6. Follow-up: Please include the paper mentioned above.

Restructure sentence - line number 192 in the follow-up section.

Abbreviations: spell check line 252.

Author Response

Answer to the Reviewers

Reviewer #1

First of all, thanks for taking the effort of reviewing the manuscript, for your kind words and for the relevant input which undoubtedly will improve the manuscript.

Comment #1

Please include comments/review about the following relevant literature that was recently published:

1)       Preterm Infant Outcomes after Randomization to Initial Resuscitation with FiO2 0.21 or 1.0. Thamrin & Oei et al. Published in J peds. Oct 2018. PMID: 30251639.

Following the reviewer’s suggestion we have commented this relevant paper which perfectly relates to the subheading 4. Target oxygen saturation during stabilization.

We have added the following comment in lines 141- 149:

Similarly, Thamrin et al., [25] reported the results of a follow up study at 2-3 years corrected age of preterm babies born at <32 weeks’ gestation and resuscitated with FiO2or either 0.21 or 1.0. They found no differences in Bayley Scales independently of the initial FiO2employed. Interestingly, in a post hoc exploratory analysis of the entire cohort, babies who did not reach SpO280% at five minutes after birth were more likely to die or have neurodevelopmental impairment (OR, 1.85; 95% CI 1.07-3.2; P=0.03). From the results of these studies [24, 25] it may be deduced that more than starting with a specific FiO2, performing an adequate ventilation and oxygen titration allowing the patient reaching a specific SpO2at a given time point after birth has a notable influence on major outcomes.

2)       Effect of various inspired oxygen concentrations on pulmonary and systemic hemodynamics and oxygenation during resuscitation in a transitioning preterm model. Chandrasekharan & Lakshminruimha et al. Published in Ped Research. Nov 2018. PMID 29967523.

The study is in a preterm lamb model where they have used term lambs as controls. Initial resuscitation with 21% O2 followed by titration of O2 led to suboptimal pulmonary vascular transition at birth in preterm lambs. Ventilation with 100% O2 in preterm lambs caused hyperoxia but reduced PVR similar to term lambs on 21% O2.

The novel part of this study is not highlighted in their abstract.

a)       The difference in systemic and pulmonary blood pressures with 100% oxygen that is not evident with 21% is startling.

b)      The oxygen delivery was higher with high oxygen use, and so was the regional cerebral saturation but not oxygen extraction.

3)       Could monitoring cerebral saturation be a better marker of oxygenation in the delivery room than peripheral saturation?

In my opinion, both are complementary. Having information of cerebral and systemic oxygenation allows to interpret the hemodynamic situation in the general and in the cerebral circulation.

4)       Would 5-minute Apgar give an idea of perfusion or should the perfusion be noted separately? 

Having SpO2 and HR at five minutes and cerebral oxygen regional saturation/FOE combined with the Apgar score gives an accurate picture of the hemodynamic situation. Moreover, together they are highly predictive of major outcomes such as death or IPVH.  There is no need for perfusion to be noted separately.

The author’s viewpoint on these hemodynamic variables and alternate methods of monitoring oxygenation will also be valuable to this review.

Cerebral Oxygen Saturation to Guide Oxygen Delivery in Preterm Neonates for the Immediate Transition after Birth: A 2-Center Randomized Controlled Pilot Feasibility Trial. Pichler & Schmölzer. Published J peds 2016. PMID: 26743498

Reduction of the burden of cerebral hypoxia during immediate transition and resuscitation after birth is feasible by crSO2 monitoring to guide respiratory and supplemental oxygen support.

Discussing the positives & negatives of the practical aspects of monitoring cerebral oxygen saturation in the delivery room would be valuable too.

Following the reviewer’s suggestion, we have also introduced a paragraph that argues that possibly, the inability of preterm infants to reach stable SpO2s after birth when ventilated with room air is in relation with the lesser vasodilatory effect of air upon the pulmonary vessels. We have briefly discussed the applicability of NIRS to optimize oxygen supplementation in the first minutes after birth including an additional reference: Pichler G et al Front Pediatr 2017.

The paragraph is between lines 127-142. It reads:

Clinical studies have shown that very preterm infants have difficulties to achieve an adequate oxygenation in the first minutes after birth when supplemented with room air. In a preterm lamb model of asphyxia/resuscitation using FiO20.21 did not reduce pulmonary vascular resistance as compared to FiO2of 1.0 [23]. On the contrary, the use of 100% oxygen caused hyperoxemia detected in the carotid arteries but didn’t cause cerebral vasoconstriction; however, cerebral oxygen extraction did not differ between groups [23]. Very preterm infants need higher FiO2than term infants to dilate the pulmonary vessels and avoid hypoxemia and/or bradycardia; notwithstanding, the risk of cerebral hyperoxia and subsequent brain damage should be avoided using intermediate inspired fractions of oxygen, closely titrating SpO2[23].  The use of near infrared spectroscopy (NIRS) in the delivery room allows to monitor evolving cerebral regional oxygen saturation and fraction of oxygen extraction rapidly and accurately after birth. Brain oxygenation measured by NIRS reaches a plateau much faster than arterial oxygen saturation. Cerebral tissue oxygenation relies not only in arterial oxygen saturation but also in cerebral blood flow, hemoglobin content and oxygen consumption. Thus, with the information provided by NIRS oxygen targeting especially in the first minutes after birth can be more accurately achieved and cerebral hyper-or-hypoxia more easily avoided [24, 25].

Comments by section:

2. Oxygen in the fetal to neonatal transition: in this section the authors mention - ‘clinical trials that are investigating the initiation of ventilation before cord clamping’. How could the research of sustained inflation in preterm infants have an impact?

To date randomized controlled trials have not proven any advantage of SLI during resuscitation compared to standard approach. The need for mechanical ventilation in the first 72 hours is reduced but primary outcomes such as death and/or BPD have not been reduced using SLI. For this reason, I haven’t added this information to this section. 

3. Initial FiO2 in the delivery room: 

How should the oxygen be titrated in the delivery room? 

We have suggested based on our experience and not on evidence-based studies that FiO2should be titrated up-and-down in 10% intervals every 30 seconds to avoid that rapid changes could alter pulmonary vessel caliber. However, in extreme circumstances changes can be performed more rapidly and with greater FiO2’s (Dawson JA et al J Pediatr 2012;160:158-61.

How soon and what should be the change while titrating oxygen? 

Changes in FiO2are highly dependent on the response of heart rate. If the baby is stable with a normal HR (>100 bpm), the team can adjust FiO2as described before assuring that the mask is in good position and air in entering the lungs.  If HR remains bradycardic, FiO2should be rapidly adjusted in greater intervals and assuring also correct mask position, and initiating intermittent positive pressure ventilation with PIP of 20-25 cmH2O. If no response then proceed to intubation.

Could the authors comment on this? Especially when there is a delay between titration and the actual oxygen concentration delivered to the lungs.

AHA mentions starting the initial oxygen anywhere from 21-30 up to 60% but not greater than 65%, while the authors have mentioned in the table (at the end of the manuscript) to start at not more than 40%. Please comment.

We are referring to ELBW infants, and our suggestion is based on the individual analysis of several hundred babies in 8 RCT in whom we registered SpO2, FiO2and HR continuously and had information on outcomes.  Our experience was that a substantial number of babies <27 weeks’ gestation initially ventilated with FiO20.21-0.3, remained bradycardic and hypoxic at 5 min after birth. However, with initial FiO2of 0.4 the number of babies remaining hypoxic or bradycardic was significantly lower, and also hyperoxia was avoided in many of them. This is a suggestion. We cannot know when a baby is going to be resuscitated which is the optimal initial FiO2but we know that using this approach we are going to be able to reach targeted saturations in most of the babies if the resuscitation procedure is correctly performed.

4. Target oxygen saturation during stabilization: 

Should we abolish minute-to-minute saturation targets and focus on a broader range in the first 5 minutes? ERC, AHA, ANZ, Finland, South Africa, Sweden all differ (this has been mentioned in the clinical practice section). 

Should there be a universal common target at 5 min? 

Is there an advantage/disadvantage in monitoring targets every minute?

Taking into consideration the results obtained in our individualized analysis of SpO2, FiO2and HR, it seems that we should aim for a HR >100 bpm at 3 minutes and SpO2> 80% at 5 minutes after birth. SpO2should be incorporated to the Apgar score.

Until we have more information, we propose a universal target SpO2of 80-85% at five minutes for any gestational age.

Monitoring every minute allows to rapidly evaluate the response to intervention by watching the tendency of HT and SpOto increase or decrease. However, excessive attention to the monitor may distract from attending the patient. Therefore, a collaborator should be in charge of this.

6. Follow-up: Please include the paper mentioned above.

Restructure sentence - line number 192 in the follow-up section.

Abbreviations: spell check line 252.

According to the reviewer’s indications we have changed the sentence in line 192. It reads now:

Results from follow up studies showed that using an initial FiO2of 1.0 increases the chance of NDI in survivors. However, when using initial FiO2of 0.6-0.65 as shown in Boronat’s paper [33] not differences in neurocognitive or sensorial outcomes were present. 

Line 252: spelling corrected.

Reviewer 2 Report

This is an excellent review of optimal inspired fraction of oxygen in the delivery room for preterm infants by Lara-Canton t al. With many reviews  / meta-analysis relating to FIO2 at resuscitation, the review needs to come out with more focus on the next step forward.  

The reviewer has the following general comments relating to the article –

The authors repeat multiple times about the need for new research. What are the shortfalls in current knowledge regarding FIO2 administered at resuscitation? How can this be addressed in the future as per the authors. What new research they would recommend to solve the current problems? Addressing research gaps methodically would help the readers on ‘what next’.

The reviewer feels that the focus is on FIO2 administration at resuscitation in this review. Is that is what the message the authors want the readers to take? SPO2 is a function of effective CPR. In short, function of heart rate and cardiac output. How would the authors relate the physiology of cardio-pulmonary resuscitation to SPO2, as a function of FIO2 administration?

Heart Rate section. The authors should discuss the physiological relationship between HR and SPO2. The question is how heart rate should be monitored in the first 5 minutes after birth? Do the authors recommend measuring HR at every minute as part of a scoring system?

HR between 60 to 100/min results in aggressive PPV management with no changes in FIO2. However, AHA guidelines suggest that increase FIO2 to 100% if HR < 60/min (with chest compression). So defining bradycardia in relation to FIO2 administration is also important.

Reference 26 – The authors have to include the reference that actually referenced hypoxia / hyperoxia..neuronal necrosis….inflammatory and pro-apoptotic pathways. The authors should credit the right work reference, as 26 is a review.

As per table - Initial FIO2 –25 6/7 GA to be started at 0.4; 28 6/7 to be started at 0.3 and 29 1/7 to be started at 0.21. However, the authors have discussed that it is the HR that is critical to determine SPO2. So, how would increasing FIO2 in these circumstances would increase SPO2?

It is reviewer’s belief that the table is not in line with AHA / ILCOR current guidelines. The authors have not explained the need for sub-stratification of infants at low GA in detail. The readers will be confused with the FIO2 to administer in the DR (there is already confusion!). There is no basis to sub stratify extremely low GA infants.

Author Response

Answers to Reviewer #2

First of all, thanks for taking the effort of reviewing the manuscript, for your kind words and for the relevant input which undoubtedly will improve the manuscript.

This is an excellent review of optimal inspired fraction of oxygen in the delivery room for preterm infants by Lara-Canton t al. With many reviews/meta-analysis relating to FIO2 at resuscitation, the review needs to come out with more focus on the next step forward.  

The reviewer has the following general comments relating to the article – 

The authors repeat multiple times about the need for new research. 

What are the shortfalls in current knowledge regarding FIO2 administered at resuscitation? How can this be addressed in the future as per the authors. 

What new research they would recommend to solve the current problems? 

Addressing research gaps methodically would help the readers on ‘what next’.

We appreciate the reviewer’s point of view and agree that criticism without pointing out solutions does not completely fulfill the aims of this review. In this regard, we have added a new section that deals with the shortfalls in our present knowledge on how to manage oxygen in the DR, and how new randomized trials that are being launched have been designed to provide strong evidence in this specific field of knowledge.

The paragraph reads as follows:

8. Seeking for an answer: forthcoming trials. 

Meta-analyses and systematic reviews performed in the last years, as shown in sections 3 and 4, have not been able to shed light on the optimal use of oxygen in preterm infants in the delivery room. Striking differences in favor of initiating resuscitation with higher (even 100%) oxygen, intermediate (60%) or lower (21-30%) have been concluded. The small size, differences in gestational ages, and varied designs and primary objectives among others, have led to different conclusions generating confusion among clinicians. To overcome this conundrum, large multicenter trials of sufficient sample size powered to look at safety outcomes such as mortality and/or major neurodevelopmental impairment are needed. The HiLo trial (NCT03825835)aims to answer if higher (60%) or lower (30%) initial oxygen concentration has an impact on patient-relevant outcomes in preterm infants of 230to 286weeks’ gestation. The null hypothesis is that neither mortality or NDI at 18-24 months corrected age will be different independently of the initial FiO2employed. The study has a cluster crossover design, unmasked randomized controlled trial comparing two oxygen concentration at the initiation of resuscitation. Hence, NICUs instead of infants will be randomized to an initial FiO2’s (e.g.: higher) until a number of babies get recruited, and then switched to the other FiO2(i.e.: lower). The analysis will be conducted using an “intention-to-treat” approach. Interestingly, SpO2will be controlled at 3 min after birth and according to the oxygen saturation increased or decreased by 20% every 60 seconds to achieve ³85% between 5-10 minutes of age. A second study, the Torpido 30/60 (ACTRN12618000879268) trial, compares initial FiO2of 0.3 vs. 0.6. This is a phase III, randomized, 2 parallel arms, non-blinded trial in preterm infants 230to 286weeks’ gestation. The aim of this study is to compare short-and-long term outcomes being the primary outcome survival free of major brain injury at 36 weeks, and secondary outcomes mortality and major brain injury. Eligible babies will be randomized to FiO2of 0.3 or 0.6 before birth. Importantly, target SpO2at five minutes should be 80-85% and at 10 minutes and after 85-95%. Both, the HiLo and Torpido 30/60 have been powered to detect an absolute risk difference around 10% in the primary endpoint. Both these ongoing trials seek to answer many of the question’s neonatologists have to face when confronted with the resuscitation of very preterm infants.

The reviewer feels that the focus is on FIO2 administration at resuscitation in this review. Is that what the message the authors want the readers to take? 

We appreciate the reviewer’s question. This is not our specific message. We have underscored several issues which we consider relevant to oxygen administration in the delivery room: (i) initial FiO2; (ii) target saturations; (iii) titration; (iv) heart rate. However, in the literature, we have a lot of information on point (i) but only recently we are getting some information and most retrospective from the rest of the points. Perhaps this is the reason why our review may seem unbalanced. We have tried to underscore the lack of information in relevant aspects of oxygen management in the DR.

SPO2 is a function of effective CPR. In short, the function of heart rate and cardiac output. How would the authors relate the physiology of cardiopulmonary resuscitation to SPO2, as a function of FIO2 administration?

Heart Rate section. The authors should discuss the physiological relationship between HR and SPO2. The question is how heart rate should be monitored in the first 5 minutes after birth? Do the authors recommend measuring HR at every minute as part of a scoring system? 

Interesting and difficult questions. I have tried to answer both in one paragraph since both are closely linked to cardiorespiratory physiology.  Oxygenation of tissue is dependent on cardiac output (stroke volume x heart rate) and hemoglobin concentration and saturation. An adequate preload and afterload of oxygenated blood provides coronary arteries with sufficient oxygen to meet the indispensable aerobic metabolism of the myocardium. The aim of resuscitation is to provide sufficient oxygen to cardiac preload to allow the pump to provide the rest of the tissue with enough oxygenated blood. Saturation is an indirect measurement of this situation, but heart rate is the cornerstone of a correct myocardial response to resuscitation maneuvers. Both should be taken into consideration, but heart rate goes first. Performing an effective ventilation should provide enough oxygen to arterial blood; however, if despite effective ventilation SpO2does not increase, FiO2should be increased as needed. 

Heart rate should be continuously monitored along the entire resuscitation process. Using pulse oximetry we lack reliable information on HR for the first 2 minutes. This fact is important because bradycardia during this sensible period of time will strongly influence the resuscitation process regarding the use of chest compressions or providing with higher oxygen concentrations, or even intubate. This is one of the arguments for using three electrodes ECG in the DR. Heart rate should be closely monitored and data registered. As has been shown, prolonged bradycardia significantly correlates with death/IVH.

We have added an initial paragraph to the Heart Rate section. It reads as follows:

Myocardial function is indispensable for adequate tissue perfusion and oxygenation. Cardiac output is the result of stroke volume and heart rate. Cardiac function is highly dependent on aerobic metabolism. Coronary blood flow is regulated by arterial blood oxygen content. Therefore, under hypoxemic conditions myocardium suffers hypoperfusion and energy exhaustion which lead to bradycardia and generalized hypoperfusion and hypoxemia [29]. Heart rate is therefore the most critical sign for an adequate response to resuscitation maneuvers.

HR between 60 to 100/min results in aggressive PPV management with no changes in FIO2. However, AHA guidelines suggest that increase FIO2 to 100% if HR < 60/min (with chest compression). So defining bradycardia in relation to FIO2 administration is also important.

Experimental studies have not shown that using 100% oxygen during bradycardia improves outcome. However, we don’t have evidence in this regard. If chest compression is initiated, at present, FiO2should be increased to 1.0.

Reference 26 – The authors have to include the reference that actually referenced hypoxia / hyperoxia..neuronal necrosis….inflammatory and pro-apoptotic pathways. The authors should credit the right work reference, as 26 is a review.

This specific review addresses all pathophysiologic mechanisms/pathways related to hypoxia and hyperoxia. We have tried to avoid an excessive number of references.

As per table - Initial FIO2 –25 6/7 GA to be started at 0.4; 28 6/7 to be started at 0.3 and 29 1/7 to be started at 0.21. However, the authors have discussed that it is the HR that is critical to determine SPO2. So, how would increasing FIO2 in these circumstances would increase SPO2?

It is the reviewer’s belief that the table is not in line with AHA / ILCOR current guidelines. The authors have not explained the need for sub-stratification of infants at low GA in detail. The readers will be confused with the FIO2 to administer in the DR (there is already confusion!). There is no basis to sub-stratify extremely low GA infants.

The table is provoking because it relates our suggestions after gathering second by second information during the first ten minutes after the birth of very preterm infants continuously monitored in the DR. 

We have evidence that HR response to oxygen after birth is closely related to the gestational age. A substantial percentage of very preterm infants do not respond to air or even 30% oxygen.  These infants can remain for 3-4 minutes in hypoxia and bradycardia, and we suspect that this could be injurious. Extremely preterm infants, especially <27 weeks are the fewer responders to lower FiO2’s and for this reason, we have suggested to initiate stabilization with 0.4 and then titrate. 

The table is not in agreement with the AHA/ILCOR current guidelines. We don’t present this information as a GUIDELINE, only as suggestions based on the individualized data of 6 published RCT. However, we agree with the reviewer that at the present stage this information could add to the existing confusion. We have modified the table accordingly.
